# Review and Perspectives of the Use of Alginate as a Polymer Matrix for Microorganisms Applied in Agro-Industry

**DOI:** 10.3390/molecules27134248

**Published:** 2022-06-30

**Authors:** Betsie Martínez-Cano, Cristian Josué Mendoza-Meneses, Juan Fernando García-Trejo, Gonzalo Macías-Bobadilla, Humberto Aguirre-Becerra, Genaro Martín Soto-Zarazúa, Ana Angélica Feregrino-Pérez

**Affiliations:** 1Faculty of Engineering, Autonomous University of Queretaro, Campus Amazcala, El Marques 76265, Mexico; bmartinez01@alumnos.uaq.mx (B.M.-C.); cmendoza27@alumnos.uaq.mx (C.J.M.-M.); fernando.garcia@uaq.mx (J.F.G.-T.); humberto.aguirreb@uaq.mx (H.A.-B.); 2Faculty of Engineering, Autonomous University of Queretaro, Cerro de las Campanas, El Marques 76010, Mexico; gonzalo.macias@uaq.mx

**Keywords:** sodium alginate, coating, encapsulation, microorganisms, biocontrol, plant-growth promotion

## Abstract

Alginate is a polysaccharide with the property of forming hydrogels, which is economic production, zero toxicity, and biocompatibility. In the agro-industry, alginate is used as a super absorbent polymer, coating seeds, fruits, and vegetables and as a carrier of bacteria and fungi as plant-growth promoters and biocontrol. The latter has a high impact on agriculture since the implementation of microorganisms in a polymer matrix improves soil quality; plant nutrition, and is functional as a preventive measure for the appearance of phytopathogenic. Additionally, it minimizes losses of foods due to wrong post-harvest handling. In this review, we provide an overview of physicochemical properties of alginate, some methods for preparation and modification of capsules and coatings, to finally describe its application in agro-industry as a matrix of plant-growth-promoting microorganisms, its effectiveness in cultivation and post-harvest, and its effect on the environment, as well as the prospects for future agro-industrial applications.

## 1. Introduction

Sodium alginate is a very versatile polysaccharide with many applications due to its high-tech functionality, it is economical to produce, and it can be obtained in bulk; it is not toxic, it has continuos quality, it is practically sterile, and it is biocompatible with microorganisms [1]. It is currently functional in the food, textile, agrotechnological, biomedical, and pharmaceutical industries since it can be easily modified through chemical and physical reactions to create matrices such as hydrogels, microspheres, microcapsules, sponges, and fibers [2].

The use of microorganisms arises as an alternative to replace the use of agrochemicals partially or totally [3], and they have been supplied in the form of inoculum, in the soil, or directly in the plant, these are preparation of beneficial microorganisms, phytostimulants, fertilizers, and biocontrol agents, without a matrix that protects them bride [4]. However, the results obtained are not as expected since these microorganisms can be negatively affected by the competition of native microorganisms or by unfavorable environmental, physical, and chemical conditions [5].

Therefore, it is relevant to implement a matrix, either in the form of spheres or film, that provides optimal conditions to improve the life of microorganisms and improve their application, to obtain the desired benefits [6], considering the economic competitiveness based on general operating costs of the formulation [7].

The purpose of this review is to know the characteristics of alginate, its applications in agriculture, its benefits, and its possible deficiencies to improve how beneficial microorganisms are applied in the soil and to provide innovative technology solutions that allow obtaining the desired results in the field.

This review article contains specific information on the use of alginate in agroindustrial processes, the different methods to obtain alginate products such as films, coatings, matrices, hydrogels, capsules, and others that contain active agents of interest to be used as fertilizers, biological controls, growth promoters, plant health preservatives, among others. The differentiator of this review is the concept of exploring a single specific compound for various processes related to agroindustrial, from plant growth and health situations to methods to extend and preserve the shelf life of fresh products.

## 2. Alginate Overview

Sodium alginate is a polysaccharide obtained from marine algae and bacteria. The chemical structure is a copolymer of blocks, which are formed from β-D-manuronic acid (M) and α-L-guluronic acid (G) linked through 1,4-glucosidic bonds (Figure 1). Its structure is heteropolymeric, that is, a combination of manuronic (M)/guluronic (G) residues, and its sequence varies according to the source from which it is obtained [8,9]. The composition, extension, and molecular weight of the sequences establish the physical properties of the alginate [10].

The main characteristic of sodium alginate is gelation with calcium ions [11]; this characteristic is responsible for its wide use in the food industry since it can be used as a thickener [12], stabilizer, and binder [13,14].

Sodium alginate has properties that allow the use of this compound in food and products, such as the capacity for water retention, gelling, as a thickener, and forming capsules and films [14]. In addition, its consumption does not present toxicity, and it is a biocompatible and human-degradable compound [15].

Together, the physical and chemical properties of the alginate determine the base of the product to be manufactured and the application method for a specific sector. As previously mentioned, the proportion of the MM, GG, and MG blocks influences the behavior of the material. A high G content provides a high gelling capacity; however, the higher the M content, the higher the viscosity. This implies that the interactions between M/G are decisive in the behavior of the gel obtained, a high ratio causes greater elasticity, while a low ratio generates weak products [16].

The strength of the interaction is another factor that is modified by the M/G ratio. This property determines the mechanical behavior of the resulting gel through the tensile strength, the breaking point, and the viscoelasticity. In addition, there are properties such as solubility, pH, and particle size that depend directly on the type of algae from which the alginate is extracted [10].

Sodium alginate can form a hydrogel, which presents a cross-linked three-dimensional network of hydrophilic polymers with a high quantity of water by ionic crosslinking and covalent crosslinking. Ionic crosslinking is the most common method to form a hydrogel and is carried out from an aqueous alginate solution with divalent cations, mainly Ca^2+^ (Figure 2) [17].

Homogeneous alginate beads are formed by the controlled introduction of divalent crosslinking cations and can be carried out by two methods: diffusion or internal configuration. The diffusion method consists of dripping a solution of sodium alginate into an aqueous solution of CaCl_2_. This process has fast kinetics and a non-homogeneous distribution of alginate, high concentration on the surface, and gradually decreases to the center of the pearl of the gel formed [18]. In the internal configuration method, the cation source is found inside the alginate solution, and its release is controlled by the pH or solubility of the cation source, and the gradual release allows the formation of gels with a homogeneous concentration of cations [19].

The main applications of alginate capsules are biomedical and pharmaceutical: to make a slow and controlled release of drugs and enzymes [20,21]; in the food industry: to microencapsulate probiotics, prebiotics [22], nutrients [23], as well as microorganisms that benefit the intestinal flora [24]; and in agriculture, for the treatment of wastewater, adsorption of heavy metals [25], for the encapsulation of bioactive substances [26], eliminate some organic pollutants [27], encapsulation of compounds to remove pathogenic bacteria [28] and to encapsulate bacteria beneficial for plant growth [29].

## 3. Formation of Alginate Coatings and Films

Sodium alginate is used as a coating, which is defined as a layer of material to maintain quality, reduce deterioration processes by microorganisms, as well as extend the shelf life of food [30]. The generation of an edible film or coating can be obtained by immersion or by spraying, by different methods such as solvent removal, microfluidization, thermal gelation, lipid solidification, casting, and electrospraying. The first technique consists of a solution with the additives, where the solvent is evaporated from the solution, resulting in a plasticizing layer of material on the food or product. Microfluidization is a process where microchannels are generated to form a network of nanoparticles that form the coating, improving the physicochemical characteristics of the food or product [31,32].

Another method is thermal gelation, which is a simple and inexpensive process, where a protein compound is subjected to heat treatment to form a structured gel as a covering for a food or product [33]. Lipid solidification is a technique for making films and coatings that consists of a mixture of components, including an oil where layers of the material are formed by immersion, which solidifies with each immersion in the mixture of components [34]. Among these methods, there is also casting; in this process it is necessary to have a previously formed layer with some method of immersion or spraying, to form an elastic and plastic layer around the food or product under controlled conditions of humidity and temperature to achieve a continuous and complete thermoforming [35].

One of the most widely used methods for its high efficiency, low cost, and versatility in the film-forming elements and coatings is electrospraying; this process has the fundamentals of spray formation. However, electromagnetism is used for training when the attraction of the solution begins between two magnetic poles. The application of a potential difference is the outside by which a continuous wire is generated while the solvent evaporates; there are variants of this technique according to the desired result so they can be capsules or fibers of different sizes from millimetric to nanometric [36,37].

### Benefits of Alginate Coatings and Films

A coating or film of sodium alginate has the particularity of being a semi-permeable layer that reduces the loss of water and solutes, in addition to keeping O_2_ and CO_2_ gases in balance controlling biochemical changes, and delays organoleptic damage such as sight and texture [38]. In addition to these characteristics as shown in Figure 3, this type of coating or film must cover some quality aspects such as versatility and ease of processing, compliance with the sensory qualities by not being noticeable in consumption, improve physicochemical properties of the feed, and regulate conditions through the addition of additives [39].

The properties of the coatings or films can be divided into two parameters to guarantee their quality and performance. The first is the permeability to water vapor, which can predict water loss; this parameter can be modified according to the composition of the coating, the interaction with the food or product, the thickness, and the flexibility of the layers. Although these last two attributes are related because the higher the thickness of the layer, the lower the flexibility. However, the thicker the coating, the greater the permeability to water vapor [40].

The mechanism of water transfer between the environment and the humidity of the food is carried out in three stages: the first stage is the condensation of the vapor that is stored in the film; the second stage is the concentration of water by activity or effect, and the third stage is evaporation or transfers to the other side of the barrier. What involves a process of equilibrium between water and the film or coating material, this balance is the adsorption and desorption mechanisms [41].

The second parameter is the mechanical properties of the coating or the film, among them are tensile strength, flexibility, stiffness, and compaction force. These properties are mainly influenced by the viscosity of the coating suspension. This is due to changes at a structural level in the composition of the alginates [30,42].

The resulting forms of a covering are shown in Figure 4, which will depend to a great extent on the relationship between the materials, which are the core and the wall, both materials must be compatible to guarantee the viability of the covering. The nature of the core or internal material determines the use of the resulting product [43].

## 4. Use of Alginate in Agro-Industry

Alginate has had different applications in agriculture in recent decades: as a matrix of microbial formulations to improve plant productivity [44], for wastewater treatments [45], as a polymer super absorbent with eliciting properties [46,47], seed coating to increase germination [15], fruit and vegetable coating to increase post-harvest quality [48], a formulation for controlled release of agrochemicals [49] and carriers of fungi and bacteria, promoters of plant growth and antagonists for biological control [7,50,51].

The use of alginate as a microbial carrier arose because conventional solid or liquid formulations have a relatively short shelf life, and the transport and storage costs are very high, without ensuring the conservation of the microbial strains [52]. The encapsulation of plant growth-promoting bacteria in polymers such as alginate is a proposed alternative in the last 30 years. Since they ensure protection and a controlled and gradual release of the inoculant in the soil, they improve the growth of plants, they have good adhesion to seeds, the material is biodegradable in the soil, and they have no toxicity. The capsules can be dried and transported easily [53]. Additionally, with a suitable method of industrial microencapsulation, it is possible to maximize yield and produce alginate-based inoculants that may be suitable for agricultural application [29]. In addition to encapsulation, microorganisms can be applied in the form of a coating, mainly antimicrobial, when the confrontation occurs on the surface of the product or food [32].

### 4.1. Biocontrol Effect through the Mechanisms of Action in Coatings

The mechanism of action of a coating can be divided into two ways: the first is the partial or total transfer of antimicrobial agents to the surface of the product or food; however, when the antimicrobial effect occurs directly on the surface or in the layers of the coating, constitutes the second form [32].

The use of sodium alginate as a coating has been used against the effect of pathogens. The encapsulation of microorganisms is an effective measure of biological control [54]. The effectiveness of fungi is due to different factors typical of their biologies, such as their life cycle, the effect on specific organisms, the production of spores, or the antagonistic activity of other fungi [55], while some bacteria produce metabolites that work for the same purpose [56].

The mechanisms for biological control by microorganisms are antibiosis, antagonism, and mycoparasitism. Table 1 shows the organisms that have been used in the study of biological control. Where each type of mechanism is used, and the type of pathogen is treated are shown.

Antibiosis is a process by which an organism generates toxic substances for another organism without direct contact between them. When the plant pathogen is a fungus, the effects produced include inhibition in sporulation, mycelial growth, and delayed activation of conidia [66]. At the cellular level, antibiosis involves a breakdown in the cell wall of pathogens, causing a loss of material it is also known that it can cause vacuolation, disintegration, and coagulation inside the cell [67]. Martínez-Padrón, et al. [64] mention the volatile and non-volatile metabolic compounds produced such as trichodermine, gliotoxin, viridine, isonitrine, and trichozianine that can produce the antibiosis effect between the antagonist and the pathogen.

The antagonistic effect is the confrontation with the action, growth, and development of another organism, by different means such as competition. Some characteristics that favor the antagonistic effect are adaptability and the capacity for development and growth. The antagonistic effect has a greater presence in the soil and the rhizosphere, where recognition between antagonist and pathogen occurs, through reactions of lecithin and carbohydrates, due to the specific action that an antagonist has on a pathogen or group of them [68,69]. The competition for nutrients is mainly for nitrogen and carbohydrates [64].

Mycoparasitism is a complex antagonist-pathogen mechanism the way it is carried out can be divided into stages. The process begins with the identification of the pathogen by the antagonist, then the interaction of the hyphae of the antagonist occurs, it is when lytic activity occurs, where enzymes that degrade the pathogen cell wall are produced, and it ends with the degradation of cytoplasmic content [70].

### 4.2. Encapsulation of Microorganisms Promoting Plant Growth and Biocontrol

Inoculation of plant growth-promoting microorganisms emerges as an alternative to replace or minimize the use of agrochemicals and clean soils affected by contamination [71], and also improves plant growth by increasing the availability of nutrients in the soil through its different functions as biostimulants, biofertilizers, and biocontrollers, leading to sustainable agricultural production and ensuring food security in a changing climate [72].

However, one of the main problems in applying bacterial inocula is that sometimes they cannot perform their specific function because the bacterial population of the inoculum progressively decreases shortly after inoculation, due to the heterogeneity and unpredictable nature of the environment, the physicochemical characteristics of the soil, and nutritional factors [7,51,73].

That is why the research is directed to the development of an inoculant formulation with a suitable microenvironment to prevent the rapid decrease in the bacterial population, during storage before use, or once it is applied to the soil [74]. In recent years, several experimental polymer-based formulations have been successful as possible bacteria-bearing matrices; these carriers offer advantages over peat by protecting bacteria from environmental stresses and gradually releasing them into the soil once the polymers are degraded by the microorganisms present in the soil [6].

One of the most used polymers in the agricultural industry is alginate, due to its physical and chemical properties, its water-holding capacity, and being a soft and non-toxic method, it allows better handling, less dust, high fluidity, high service life, less abrasion and better soil establishment [75]. Furthermore, its degradation products have been found to influence the plant’s physiological activities as elicitors, promote germination, elongation of shoots, and increase root growth [76,77].

How this polymer is used as a matrix of bacterial inocula is encapsulation, which is a very versatile technology that protects bacteria from biotic and abiotic factors since it provides a beneficial physical barrier [78]; also, it can be modified with nutrients to improve the short-term survival of bacteria after inoculation [79]. Furthermore, encapsulation allows bacteria to be released into the target medium in a slow and controlled manner, thus increasing long-term effectiveness without losing their ability to stimulate plant growth [80]. Table 2 shows some microorganisms that have been encapsulated in alginate-based formulations for agricultural purposes.

### 4.3. The Potential Effect of Alginate Encapsulation on Plant Growth

Fages [91] began using alginate to encapsulate beneficial bacteria from *Azospirillum brasilense* and *Pseudomonas fluorescens*. His experiment was successful in wheat plants under field conditions; he found that polymer-protected bacteria survive in the field long enough to have a beneficial effect on plants; in addition, the colonization in roots by the bacteria released in alginate capsules was higher than that obtained by direct inoculation. These results provide strong evidence of the efficiency of alginate encapsulation in achieving the controlled release and protection of bacteria from the environment. From that moment, different investigations to obtain a suitable formulation for different types of plant growth-promoting bacteria at a low cost [6].

A study where the bacteria *Methylobacterium oryzae* and *Methylobacterium suomiense* co-added with *Azospirillum brasilense* were encapsulated in alginate shows that, from an initial concentration of viable cells of 10^9^ CFU/g, after 12 months at room temperature 10^8^ CFU/g of *M. oryzae* and 10^6^ CFU/g of *M. suomiense* were conserved, which conferred stress reduction in inoculated tomato plants [87]. Likewise, the immobilization of *Pseudomonas plecoglossicida* in sodium alginate promotes the symbiotic development of an arbuscular mycorrhizal fungus in potato seedlings [111].

Although alginate has many advantages to be used as a matrix to encapsulate bacterial inocula, it has a relevant limitation: the loss of bacteria during the preparation of the capsules, and also the presence of macrospores in the alginate matrix facilitates the diffusion of hydrophilic molecules. Therefore, the admixture of a filling material in the alginate matrix, such as starch, is a profitable solution strategy, since this material is abundant, cheap, renewable, and fully biodegradable [80].

Some studies show that encapsulation with both compounds improves the properties of the capsules. Additionally, the degradation and release can be controlled by changing the amount of addition of the components [124]. Wu, et al. [116] developed biodegradable and controlled release formulations of the *Raoultella planticola* bacteria, and they determined that the properties of the capsules can be modulated by varying the amounts of starch and alginate, without the bacteria losing their properties to improve plant growth, which has application to meet the needs of agricultural production.

For their part, He, et al. [80], carried out a study to evaluate the survival and colonization efficiency of *Pseudomonas putida* encapsulated in a sodium alginate-based formulation, on cotton plants under saline conditions. They found that the survival rates were 89.67, almost 9% higher than in the free cells. Additionally, on day 49 of the experiment, an increase was observed in the encapsulated bacterial population. Regarding cotton, an increase in biomass was found for the encapsulated strain, which can be attributed to the increase in the number of bacteria and the high production of indole-3-acetic acid and gibberellin, which is why microencapsulation with sodium alginate, inexpensive starch, and bentonite is recommended.

In addition to starch, sodium bentonite has been used in combination with alginate to develop effective biofertilizer formulations that minimize production costs. The mixture encapsulation efficiency is almost 100, 88.9% of *Raoultella planticola* bacteria survived after 6 months of storage and swelling, biodegradability, and release rate were found to increase with increasing alginate content, presenting a first-order release, which proves a slow-release, ideal for farmland [115].

Previous studies show that alginate is a very useful polymer for the encapsulation of bacteria. However, there are some experiments with contradictory results. Those indicate that when encapsulating bacteria in sodium alginate, populations are diminished due to inert carrier, which also minimizes bacterial viability. Additionally, the irregular surface of the capsules can negatively influence the release of bacteria, which reduces the colony-forming units present in the soil and the roots of plants, modifying inoculum effectiveness [125].

In a study by Bashan, et al. [126], it was demonstrated that the alginate capsule structure has low mechanical resistance, which produces an unstable and uncontrolled bacterial release. In addition, the cellular mortality during the drying of the capsules is a critical point to improve. Without losing sight of the fact that the large-scale production of alginate capsules and their application in the field are still limited, in addition to the fact that the cost of production is relatively high [7].

### 4.4. Alginate Capsule Shelf Life and Microbial Viability

One of the main reasons why the use of a matrix to contain plant growth promoter and biocontrol microorganisms is implemented is to increase their viability and survival; that is, to allow microorganisms introduced into the soil or plant to have a greater resistance to biotic and abiotic factors that can decrease its viability. Therefore, the evaluation of the survival of the microorganisms in preparation based on a matrix or carrier in storage for a certain period is essential to decide on suitability [101]. Furthermore, it has been established that a quality formulation must supply enough cells for effective colonization of the plant rhizosphere to improve plant growth [127].

Some studies suggest that the survival of alginate-encapsulated microbial cells is mainly related to the type of microorganism used [1], as is the case of *Pseudomonas corrugata*, deformation, and degradation of capsules, which can be attributed to the release of certain acids by the bacteria [101]. Therefore, the use of alginate will depend on the type of microorganism to be encapsulated, it is necessary to know its physical–chemical properties and the nature of the compounds it releases.

The viability and survival of microorganisms are also affected by compounds added as additives. The use of calcium gluconate in alginate capsules as a matrix of the fungi *Metarhizium brunneum* and *Saccharomyces cerevisiae* improves the viability after drying and rehydration, the hygroscopic properties, the shelf life, and the supply of nutrients, for which it is recommended to use this compound to increase microbial viability after drying, which is a crucial step for the survival of encapsulated microorganisms [128].

In a study by Trivedi and Pandey [101], they measured the survival rate over time to determine the ability of the formulations to improve the survival of *B. subtilis* and *P. corrugata*, which, being in alginate capsules was higher when compared to a char or broth matrix. They also found that the bacterial population in alginate capsules was above 10^6^ CFU/g after three years, suggesting that plant growth-promoting bacteria may survive in capsules of alginate for long periods.

Furthermore, alginate capsules can stably preserve *Mesorhizobium ciceri* and *Bradyrhizobium japonicum* during long storage periods, longer than one year, maintaining a stable concentration of colony-forming units [129].

The viability and survival of the cells encapsulated in alginate depend on the characteristics of the microorganism, the release of compounds that damage the structure of the capsule, and the compounds that are used as additives. These characteristics will affect the quality of the capsule and the time it will remain in good condition. However, according to the studies indicated above, alginate can be considered a viable alternative to preserve microorganisms in storage for approximately one year, but having an adequate estimation is necessary to evaluate cell viability with the microbial strain under storage conditions.

### 4.5. Environmental Evaluation after the Application of Encapsulated Microorganisms

The microbial and fermentative activity of the soil is an important factor in the hydrolytic and degradation processes of alginate capsules. It is well known that there are many microorganisms that degrade natural polymers and use them as a growth substrate [130], in addition to other substances present in the soil that bind to Ca^2+^ ions and also cause the degradation of the alginate capsule [131].

A biodegradation process study by Shcherbakova et al. [129] shows that the presence of alginate capsules in the soil activates microorganisms, making the mineralization processes of organic matter more efficient compared to soils without alginate, increases the total number of soil microorganisms, increases hydrolytic and oligotrophic microorganisms, which indicates that the processes of destruction of organic substances increase. Therefore, soil microorganisms actively react to the introduction of alginate, breaking it down and using it as an additional nutritional substrate. The changes in the capsules are observed 7 days after the addition with a reduction in size; however, until 14 days it is indicated that the degradation process begins.

Exudates from the roots of plants rich in organic acids stimulate the multiplication of microorganisms present in the rhizosphere and accelerate the destruction of the polymer matrix, facilitating the release of encapsulated microorganisms into the environment. The increase in the number of microorganisms coincides with the period of active growth and development of the radicle of the plants; therefore, the introduction of microorganisms in alginate provides a controlled release and retains them for successful colonization, preventing their death before they manage to colonize the root of the plant or vegetable [132].

Belaid, et al. [133] evaluated the survival and proliferation of *Azospirillum brasilense* immobilized in alginate in the soil at different humidity levels, and they found that the strain survived for a period of 75 days in the soil without plant roots, being relevant humidity and the amount of organic matter, without negative interaction effect with native microbial strains.

In addition to the release and colonization capacity of the encapsulated microorganisms, another important aspect to consider is that the characteristics of the strain are conserved. Trivedi and Pandey [101] made qualitative estimates for the characteristics related to plant growth promotion and biocontrol of bacteria *B. subtilis* and *P. corrugata* recovered from the alginate capsules and found that the bacteria encapsulated in alginate positively affected the growth parameters of wheat, increased colonization capacity, and fresh root weight when comparing the results with the broth-based formulation.

The ability of alginate-encapsulated *Pseudomonas fluorescens* to produce 2,4-diacetylfloroglucinol, an antifungal metabolite, is not affected after 12 months in storage at 4 °C and 28 °C ± 2 °C. In addition, the bacteria showed colonization of effective roots and protection against pathogenic fungi in sugar beets [109]. Encapsulation of *Mesorhizobium ciceri* and *Bradyrhizobium japonicum* increases the number of nodules in chickpea and soybean roots, as well as the weight of the nodules compared to non-encapsulated bacteria [129].

Likewise, a study shows that the inoculation of wheat plants with bacteria recovered from dry pearls after 14 years of storage of *Azospirillum brasilense* and *Pseudomonas fluorescens* had the same effect on colonization and increase in plant growth when compared with contemporary cultivated strains [92].

As well as these, there are many studies that demonstrate the effectiveness of encapsulation with alginate when applied in the soil, preserving, and sometimes increasing the effect of encapsulated microorganisms as promoters of plant growth and biocontrol. This considers the alginate as an alternative for inoculation of microorganisms into the soil without losing effectiveness in the final product. However, it is necessary to make pertinent evaluations for each microbial strain.

In resume, the encapsulation of plant growth-promoting microorganisms has several advantages compared to cell-free formulations. The encapsulation allows a gradual release, improves the physiological activity of microorganisms, reduces the risk of contamination in storage and transport, and significantly increases cell viability due to protection against adverse environmental factors. Encapsulated products can be stored over a wide temperature range. However, some encapsulation techniques are hard to scale industrially and are expensive.

### 4.6. The Economic Aspect of the Use of Alginate in Agriculture

Sodium alginate is a widely used biopolymer due to its biocompatibility, biodegradability, ability to form hydrogels, and water-solubility properties. Currently, the demand for the manufacture of alginates is expected to enhance due to its current and future biomedical, bioengineering, and food applications. As mentioned above, alginate can be obtained from numerous brown algae. A large amount of biomass of the brown alga *Sargassum horneri*, known as “golden tide” has recently caused a severe ecological impact on coastal ecosystems in many countries [134]. Massive algal blooms are a menace to the economy and tourist attractions of affected countries. Therefore, governments and researchers are exploring the uses of *S. horneri* to obtain natural products and useful materials for the golden tides’ control and sustainable management.

The principal industrial use of marine macroalgal residues is related to the extraction and purification of the polysaccharide fraction, which can range between 4 and 76% dry weight, being the alginate in brown algae. The extraction of alginate in the industry based on marine algae residues is well examined, and following an adequate recovery route, compounds with a higher value and a lower production cost can be obtained [135]. A study by Fernando, et al. [136] suggests extracting alginate from *S. horneri* to produce alginate microparticles for drug delivery. The previous study is aligned with the principles of circular economy, where a residue, in this case, the excess coastal biomass of brown algae, can be used as a resource.

Microencapsulation and film formation with sodium alginate as a vehicle for plant-growth-promoting and biocontrol microorganisms can help to prolong their useful life, facilitate their incorporation into agricultural systems and allow controlled release. The objective of having a controlled release is to reduce the amount of product that is added to the soil, which decreases operating costs and ensures the implementation of a constant and correct amount of the bioactive. Therefore, the product is not released into the environment, avoiding environmental problems [137].

There are different methods for microencapsulation with alginate: extrusion, which is easy to apply and industrialize but is not compatible with thermosensitive microorganisms; coextrusion, which is a process for a large payload but has high costs; spray-drying, it is an economical technology and equipment is widely available, it has good encapsulation efficiency, and it is a fast process, but it has a low production volume; spray-cooling is an economical technology for encapsulation but requires high energy and a long process time, which makes it more expensive; and complex coacervation, it has a high payload and no specific equipment is required; however, it is a costly process due to the complexity of the technique [138].

In general, laboratory-scale processes for microencapsulation and alginate film formation are hard to scale industrially; however, equipment such as electrostatic extruders, mechanical jet cutters, and impact spray method, among others, are currently available. To assess whether an industrial process is techno-economically viable, it is essential to carry out an analysis that includes the mass and energy balance around the unit operations included in the process and add an accounting of the costs and potential income associated with the entire installation at industrial scale.

Strobe, et al. [139] performed a techno-economic analysis to evaluate two industrial encapsulation techniques with sodium alginate: a traditional external gelation process and a process that achieves in situ alginate cross-linking during spray drying. They found that the cross-linking alginate microcapsule process exhibited lower investment and annual operating costs than the external gelation process, and the process required less water and utility usage. Dimitrellou, et al. [140] used the extrusion technique for the encapsulation of *Lactobacillus casei* and, according to their analysis of production costs, they determined that the introduction of the encapsulation is profitable if an increase in the sale prices of the new product range between 0.04 and 0.05 €/L, depending on the volume of production.

In economic terms, the most important thing is to choose the encapsulation technique or film production according to cost–benefit, which allows obtaining high efficiency, with few steps in the process at a low cost. The application of the different products in the agroindustry reduces the amount of product to be used (biofertilizer, biostimulant, biocontrol), operating costs and minimizes the environmental damage caused by the excessive use of agrochemicals.

### 4.7. Technology Used in the Large-Scale Production of Alginate Products in Agriculture

The large-scale production of alginate-derived products is influenced by the conditions of the alginate used, the nature of the model used in the products, and the technology used for its preparation. Cross-linked alginate is used in the production of microcapsules and hydrogels on a large scale due to the advantages presented during the production process. The main advantage is the simplification of unit operations during gelation and drying that decreases the resources used and therefore, the cost of investment and operating expenses are reduced [139].

The implementation of different unit operations depends on the type of model retained in the alginate products, these compounds can be enzymes, concentrates, bacteria, vitamins, and oils, among others [141]. Therefore, the nature of the additive or active agent added to the alginate products modifies the stages of the production process. Thus, a hydrophilic material requires fewer unit operations, unlike a hydrophobic material that requires a prior emulsification stage [139].

Emulsion technology is the most efficient method for the large-scale production of viable bacteria in alginate products for use in agro-industrial processes, this is due to the absence of heat sources in the method that damage the active agent [142]. However, the spray-drying technology to obtain microcapsules and particles enriched with the active agent has been reported to lower production costs and higher quality in terms of the content retained in the alginate product; therefore, it is the technology best suited for industrial production [143]. In addition, there are other techniques such as extrusion that are more expensive and difficult to scale to higher production even when the model of additive or active agent is microorganisms [144].

Currently, alginate products added with retained models for agroindustrial use are marketed mainly in fertilizers, the most common uses are fertilizers of plant extracts, leachates, fungi, and bacteria that promote plant health [145].

## 5. Perspectives for the Use of Alginate in Agro-Industry

Alginate is considered a very useful polymer in the agro-industry due to its great versatility and qualities, zero toxicity, and compatibility with many compounds; it has been used with wide benefits as a matrix for beneficial microorganisms. In the form of capsules to be implemented during cultivation obtaining improvements in yield, quality, and pest control, thus promoting sustainable agriculture practices, which contribute to increased agricultural productivity, decreased use of agrochemicals, and less environmental impact [146]. In addition, in the form of a coating, to reduce contamination by pathogenic microorganisms and improve post-harvest quality [32]. However, there are still unsolved problems, so a future frontier in the field of microbial encapsulation and coatings is the development of matrices with polymeric nanoparticles as an additive or microencapsulated formulation, to solve the main problems of technology and improve the stability of microorganisms concerning environmental conditions.

The implementation of low-cost additives or substances necessary for the inoculum can be beneficial to increase the useful life of the product or control the release profile of microorganisms [147,148]. With this, there is also the need to carry out a more in-depth analysis of the alginate–additives–microorganisms–soil–plant system relationship; to obtain important information that allows us to understand the functional characteristics of the encapsulated microorganisms and establish strategies for their application [7]. On the other hand, in the preparation of microbial coatings for food, it is necessary to make a complete analysis of the effect that the coating has on the quality of food and its by-products, as well as on the consumer. The previous is to understand the mechanisms of action of microbial inoculate, not only to contain contamination by pathogens but also their relationship with food and with the final consumer.

The future trends are directed towards the development of microenvironmental conditions to facilitate the reproduction, growth, and functional activities of microorganisms encapsulated or disposed of in the coating. Finally, it is imperative to do a comprehensive assessment of environmental and health safety issues before the technology is moved to the industrial level, to provide complex, efficient, safe, economically acceptable, and easy-to-apply biotech products that ensure health and plant growth as well as consumer acceptability.

Another perspective for alginate application is the encapsulation of microalgae. This technology increases biomass and protects the microalgae from predators and toxic contaminants, which reduces production costs. The application of encapsulated microalgae can be used in different areas of the agro-industry. Dębowski, et al. [149] found that by immobilizing and encapsulating *Chlorella vulgaris* it is possible to increase biomass, and CO_2_ sequestration also increases. On the other hand, Cheirsilp, et al. [150] found that immobilizing the microalgae *Nannochloropsis* sp. in alginate beads for use in phytoremediation of industrial effluents significant increases biomass and lipid production, as well as nitrogen and phosphorus removal and CO_2_ mitigation. Therefore, it is relevant to increase research on the immobilization of microalgae in sodium alginate since it has multiple applications for the removal of pollutants and wastewater treatment.

In response to the need to improve the implementation of microorganisms that promote plant growth and biocontrol in agriculture, our work team proposes using alginate as a matrix and adding compounds that allow improving the physicochemical characteristics of capsules and coatings; through multidisciplinary studies that allow evaluating the viability and effectiveness from the ecological, biological, technological, and economic aspects, to obtain a product that can be transferred to the industrial level and subsequently used successfully by agricultural producers.

## 6. Conclusions

Alginate is a polymer that has allowed significant progress in the technology of formulations for microorganisms that promote plant growth and biocontrol due to its physicochemical properties, biocompatibility, and non-toxicity. Likewise, some studies suggest that the use of certain additives positively affects its protection capacity, increasing microbial viability, and survival, perceived as a beneficial effect on plant growth and biological control. However, in some cases, some compounds have been shown to exert a negative effect on the microbial encapsulation system and coatings; therefore, it is necessary to analyze the relationship between alginate, additives, and microorganisms to be implemented. As well as the effect they have on the soil and plant, food, and consumer systems to obtain the necessary information to propose a quality formulation, safe, effective, and easy to apply both for agricultural producers and for workers in the food industry, which confers benefits on crop quality and health.

## Figures and Tables

**Figure 1 molecules-27-04248-f001:**
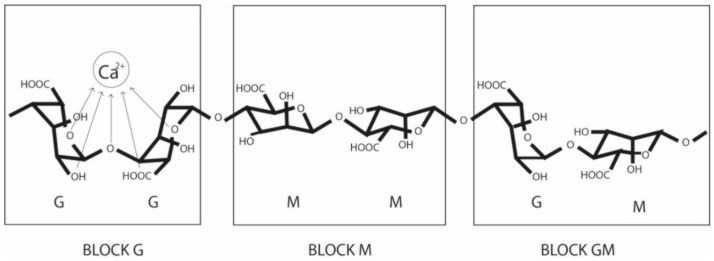
Chemical structure of alginate differentiating G-blocks and M-blocks. The selective binding of G-block with divalent cations such as calcium is represented, which produces the formation of a hydrogel.

**Figure 2 molecules-27-04248-f002:**
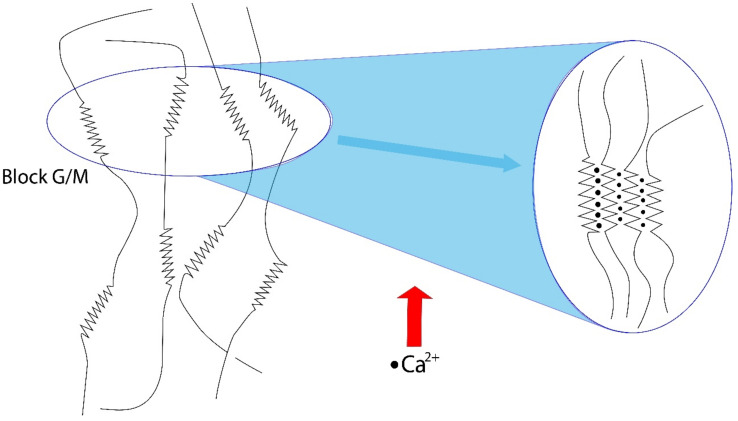
Alginate egg box fix in the presence of Ca^2+^ ions.

**Figure 3 molecules-27-04248-f003:**
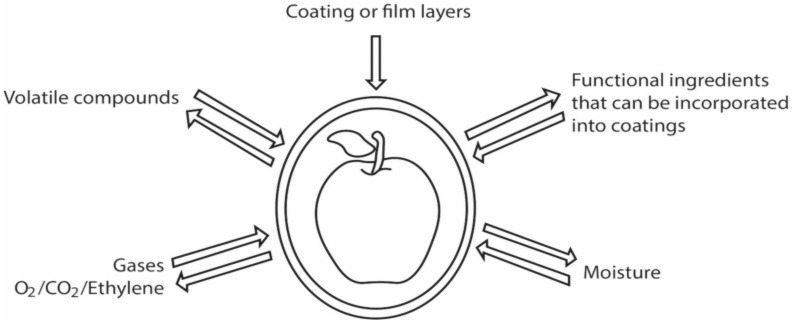
Types of transfers controlled by edible barriers in food.

**Figure 4 molecules-27-04248-f004:**
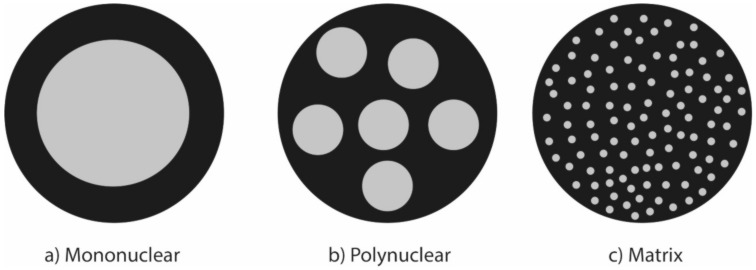
Structures resulting from a coating. (**a**) mononuclear: a single cluster of material within the capsule; (**b**) polynuclear: different large cluster of material within the capsule; (**c**) matrix: small centers or dispersed material within the capsules.

**Table 1 molecules-27-04248-t001:** Microorganisms used for the biocontrol of pathogens.

Microorganism (Antagonistic)	Pathogen or Causative Agent	Host	Mechanism of Action	Reference
*Aureimonas altamirensis* *Bacillus toyonensis* *Herbaspirillum huttiense* *Micronospora marítima*	*Phytophthora nicotianae*	Pineapple	Antagonism	[57]
*Bacillus amyloliquefaciens*	*Burkholderia glumae*	Rice	Antibiosis	[58]
*Bacillus amyloliquefaciens*	*Penicillium italicum*	Lime	Antagonism	[14]
*Bacillus subtilis*	*Penicillium digitatum*	Orange		
*Cryptococcus diffluens*	*Fusarium oxysporum*	Chilli pepper		
*Debaryomyces hansenii*	*Alternaria solani*	Tomato		
*Rhodotorula minuta*	*Neoscytalidium dimidiatum*	Fig tree		
*Stenotrophomonas rhizophila*	*Colletotrichum gloeosporioides*	Papaya		
	*Alternaria alternata*	Basil		
	*Fusarium solani*	Chickpea		
	*Curvularia* sp.	Palm		
*Bacillus aryabhattai*	*Burkholderia glumae*	Rice	Antibiosis	[59]
*Burkholderia vietnamiensis*				
*Bacillus foraminis*	*Alternaria alternata*	Tomato	Antagonism	[60]
*Bacillus subtilis*	*Corynespora cassiicola*			
*Bacillus thuringiensis*	*Stemphylium lycopersici*			
*Bacillus thioparans*				
*Micrococcus yunnanensis*				
*Paenibacillus polymyxa*				
*Bacillus subtilis*	*Pythium aphanidermatum*	Cucumber	Mycoparasitism	[61]
*Trichoderma asperellum*				
*Trichoderma fertile*				
*Beauveria bassiana*	*Diaphorina citri*	Citrus	Mycoparasitism	[62]
*Hirsutella citriformis*				
*Isaria javanica*				
*Simplicillium lanosoniveum*				
*Rhodotorula mucilaginosa*	*Botrytis cinérea*	Roses	Antagonism	[63]
*Trichoderma asperellum*	*Phytophthora capsici*	In vitro	Non-volatile metabolites	[64]
*Trichoderma hamatum*				
*Trichoderma harzianum*	*Zymoseptoria tritici*	Bean	Mycoparasitism	[65]

**Table 2 molecules-27-04248-t002:** Plant growth-promoting microorganisms encapsulated in alginate-based formulations to increase microbial viability and efficacy tested in different plants and vegetables.

Microorganism	Formulation Material	Plant	Reference
*Azospirillum brasilense*	Alginate and skimmed milk	Wheat	[81]
	Alginate	Tomate	[82]
	Alginate	Desert trees	[83]
	Alginate and starch	In vitro	[84]
*Azospirillum lipoferum*	Alginate	Corn	[85]
	Alginate and humic acid	Rice	[86]
*Azospirillum* sp. and *Methylobacterium* sp.	Alginate	Tomato	[87]
*A. brasilense* and *Bacillus pumilus*	Alginate	Legume trees	[88]
*A. brasilense* and *Chlorella sorokiniana*	Alginate	Tomato	[89]
	Alginate	Sorghum	[83]
*A. brasilense* and *Pantoea dispersa*	Alginate and organic olive waste	*Pinus halepensis*	[90]
*A. brasilense* and *P. fluorescens*	Alginate	In vitro	[91]
	Alginate and skimmed milk	Wheat	[92]
*A. lipoferum, Bacillus polymyxa,* and *Nostoc muscorum*	Alginate, carboxymethyl cellulose, and talc	Bread wheat	[93]
Nitrogen-fixing bacteria	Alginate y maltodextrin	In vitro	[94]
*Bacillus megaterium*	Alginate	Corn	[95]
	Alginate and humic acid	Rice	[96]
*Bacillus subtilis*	Alginate and pea protein	*Brachypodium distachyon* and *Phleum pretense*	[97]
	Alginate and jelly	In vitro	[98]
	Alginate and humic acid	Lettuce	[99]
*Beauveria bassiana*	Alginate and wheat bran	Cattle pasture	[100]
*B. subtilis* and *Pseudomonas corrugata*	Alginate	Wheat	[101]
	Alginate and skimmed milk	Corn	[102]
*Enterobacter* sp.	Alginate and skimmed milk	Lettuce	[103]
*Glomus desertícola (AM mycorrhizae)*	Alginate	Tomato	[104]
*Klebsiella oxytoca*	Alginate	Cotton seeds	[105]
*Methylobacterium oryzae, Methylobacterium suomiense* y *Azospirillum brasilense*	Alginate	Tomato	[87]
*Pantoea agglomerans*	Alginate, glycerol, and chitin	In vitro	[106]
*Pseudomonas* sp.	Alginate and attapulgite	In vitro	[107]
*Pseudomonas fluorescens*	Alginate, skimmed milk, and clay	Wheat	[108]
	Alginate	Sugar cane	[109]
	Alginate	Corn	[110]
*Pseudomonas plecoglossicida*	Alginate	Potato	[111]
*Pseudomonas putida*	Alginate	Corn	[112]
*Pseudomonas striata*	Alginate	In vitro	[113]
*Pseudomonas putida* y *B. subtilis*	Alginate and humic acid	Lettuce	[114]
*Raoultella planticola*	Alginate and bentonite	In vitro	[115]
	Alginate, chitin and bran	Cotton	[116]
*Raoultella terrigena*	Alginate and starch	In vitro	[84]
*Rhizobium* spp.	Alginate	*Leucaena leucocephala*	[117]
*Serratia marcescens*	Alginate	Corn	[118]
*Streptomyces* sp.	Alginate, starch, and talc	Tomato	[119]
*Trichoderma asperellum*	Alginate	In vitro	[120]
*Trichoderma harzianum*	Alginate and chitosan	In vitro	[121]
*Trichoderma viride*	Chitosan and alginate	In vitro	[122]
*Yarrowia lipolytica*	Alginate	Bean	[123]

## Data Availability

Not applicable.

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
