# Peer review of "Review and Perspectives of the Use of Alginate as a Polymer Matrix for Microorganisms Applied in Agro-Industry"

_molecules, 2022, doi:10.3390/molecules27134248_

Round 1
Reviewer 1 Report
The manuscript ID molecules-1784838 entitled “Review and perspectives of the use of alginate as a polymer matrix for microorganisms applied in agro-industry” is an interesting and valuable study. The Authors carefully analysed the available literature summarized the state of the art of alginate and its applications in agriculture. The manuscript is well written and the layout is correct and logical. The authors have actually separated the chapters. In my opinion, this is a fairly well-read study. It fully corresponds to the profile of the Molecules journal. The Authors properly present the current scientific achievements in the presented field. The use of 135 references in the manuscript is very impressive.
My suggestions below:
1. A graphical abstract would be very useful for the promotion of the manuscript and the understanding of the authors' intentions by the readers.
2. The authors properly presented the technological aspects of the production of alginate carriers. However, the manuscript does not present the economic aspect of this solution. How much does it cost, is it economically effective, is it in line with the principles of the circular economy and is the technology environmentally friendly?
3. This is a very important aspect that has been completely ignored by the Authors. Please provide basic economic data.
4. Please describe, present and evaluate the strengths and weaknesses production of the characterized products. An interesting issue would be the evaluation of technological and economic competitiveness of such solutions.
5. Please, present whether this type of technology is used on a large scale, or whether there are companies producing it on a technical scale.
6. In Chapter 5, Perspectives for the use of alginate in agro-industry, the possibility of immobilizing microalgae in alginate carriers and their use in the protection of the environment and agroidustry should be considered: https://doi.org/10.3390/atmos12081031, https: // doi. org / 10.1016 / j.biortech.2009.02.076, https://doi.org/10.1016/j.biortech.2017.06.016.
7. Summing up, the manuscript is an interesting study that meets the standards of review work. I believe that after introducing the suggested changes and additions, it may be published.
Author Response
Dear Review
We appreciate all the comments received from the reviewers, which were addressed and described in a timely manner below. We are sure that these observations substantially enrich the manuscript with ID: molecules-1784838 entitled "Review and perspectives of the use of alginate as a polymer matrix for microorganisms applied in agro-industry".
Each of the observations made by the reviewers are addressed below:
Review 1
The manuscript ID molecules-1784838 entitled “Review and perspectives of the use of alginate as a polymer matrix for microorganisms applied in agro-industry” is an interesting and valuable study. The Authors carefully analysed the available literature summarized the state of the art of alginate and its applications in agriculture. The manuscript is well written and the layout is correct and logical. The authors have actually separated the chapters. In my opinion, this is a fairly well-read study. It fully corresponds to the profile of the Molecules journal. The Authors properly present the current scientific achievements in the presented field. The use of 135 references in the manuscript is very impressive.
My suggestions below:
- A graphical abstract would be very useful for the promotion of the manuscript and the understanding of the authors' intentions by the readers.
Line 29-30. We include the graphical abstract with the main idea of the review. The file is also attached separately.
- The authors properly presented the technological aspects of the production of alginate carriers. However, the manuscript does not present the economic aspect of this solution. How much does it cost, is it economically effective, is it in line with the principles of the circular economy and is the technology environmentally friendly?
Line 430-489. We include a subtitle to cover the economic aspect of the application of alginate in agribusiness; here, we describe in detail the economic and technological advantages and disadvantages of the different techniques and processes for encapsulation without neglecting the ecological aspect and from the point of view of the circular economy.
- This is a very important aspect that has been completely ignored by the Authors. Please provide basic economic data.
Line 430-489. These lines cover the economic aspect of sodium alginate use as a vehicle for microorganisms that promote plant growth, and biocontrol, from obtaining the alginate, encapsulation process and uses in agroindustry.
- Please describe, present and evaluate the strengths and weaknesses production of the characterized products. An interesting issue would be the evaluation of technological and economic competitiveness of such solutions.
Line 422-428. We add summary information about the main advantages, and disadvantages of plant growth-promoting microorganisms' encapsulation.
- Please, present whether this type of technology is used on a large scale, or whether there are companies producing it on a technical scale.
Line 490-515. We added a section containing information on the technology used in the large-scale production of alginate products in agriculture.
- In Chapter 5, Perspectives for the use of alginate in agro-industry, the possibility of immobilizing microalgae in alginate carriers and their use in the protection of the environment and agroidustry should be considered: https://doi.org/10.3390/atmos12081031, https: // doi. org / 10.1016 / j.biortech.2009.02.076, https://doi.org/10.1016/j.biortech.2017.06.016.
Line 547-557. We include a paragraph to emphasize the application of microalgae encapsulation in alginate for agro-industrial purposes: the removal of pollutants and wastewater treatment.
Summing up, the manuscript is an interesting study that meets the standards of review work. I believe that after introducing the suggested changes and additions, it may be published.
Reviewer 2 Report
The review of Angélica Feregrino-Pérez and coauthors is about sodium alginate and its perspectives of the use as a matrix for microorganisms. Review contains 135 references in my opinion not always relevant (the novelty of the links could be higher). Figures were redrawn by the authors from the reviewed original articles.
I did not find the physicochemical characteristics of sodium alginate, which was stated in the abstract.
The scientific novelty of this review is questionable (Scopus or Google Scholar contains more than a thousand reviews on a request “sodium alginate microorganism coating” in 2020-2022). The quality of this review is below existing ones:
“Encapsulation efficiency and survival of plant growth-promoting microorganisms in an alginate-based matrix – A systematic review and protocol for a practical approach” 10.1016/j.indcrop.2022.114846
“Seaweed derived alginate, agar, and carrageenan based edible coatings and films for the food industry: a review” 10.1007/s11694-021-01277-y
However, I didn’t find plagiarism, critical flaws or smth bad to reject this review. I recommend to add physicochemical characteristics of alginate and to highlight how the review of the authors differs from the thousands of existing ones
Author Response
Dear Review
We appreciate all the comments received from the reviewers, which were addressed and described in a timely manner below. We are sure that these observations substantially enrich the manuscript with ID: molecules-1784838 entitled "Review and perspectives of the use of alginate as a polymer matrix for microorganisms applied in agro-industry".
Each of the observations made by the reviewers are addressed below:
Review 2
The review of Angélica Feregrino-Pérez and coauthors is about sodium alginate and its perspectives of the use as a matrix for microorganisms. Review contains 135 references in my opinion not always relevant (the novelty of the links could be higher). Figures were redrawn by the authors from the reviewed original articles.
I did not find the physicochemical characteristics of sodium alginate, which was stated in the abstract.
Line 62-117. We describe the chemical structure of alginate since it is a crucial point to understand the behavior of its physicochemical properties. we add from line 82-93 more information about the properties and their interaction.
The scientific novelty of this review is questionable (Scopus or Google Scholar contains more than a thousand reviews on a request “sodium alginate microorganism coating” in 2020-2022). The quality of this review is below existing ones:
“Encapsulation efficiency and survival of plant growth-promoting microorganisms in an alginate-based matrix – A systematic review and protocol for a practical approach” 10.1016/j.indcrop.2022.114846
“Seaweed derived alginate, agar, and carrageenan based edible coatings and films for the food industry: a review” 10.1007/s11694-021-01277-y
However, I didn’t find plagiarism, critical flaws or smth bad to reject this review. I recommend to add physicochemical characteristics of alginate and to highlight how the review of the authors differs from the thousands of existing ones.
Line 55-61. We add the requested information on the differentiation of this review work with others published.
Round 2
Reviewer 1 Report
Manuscript has been corrected agree with my remarks and sugestions. In my opinion this is valuable and interesting work. It can be publish in current form.